# Evaluation of Non-Invasive Gargle Lavage Sampling for the Detection of SARS-CoV-2 Using rRT-PCR or Antigen Assay

**DOI:** 10.3390/v14122829

**Published:** 2022-12-19

**Authors:** Ondrej Bouska, Hana Jaworek, Vladimira Koudelakova, Katerina Kubanova, Petr Dzubak, Rastislav Slavkovsky, Branislav Siska, Petr Pavlis, Jana Vrbkova, Marian Hajduch

**Affiliations:** 1Institute of Molecular and Translational Medicine, Faculty of Medicine and Dentistry, Palacky University, 779 00 Olomouc, Czech Republic; 2Laboratory of Experimental Medicine, University Hospital Olomouc, 775 15 Olomouc, Czech Republic; 3Cancer Research Czech Republic, 779 00 Olomouc, Czech Republic

**Keywords:** SARS-CoV-2, self-sampling, gargle lavage, PCR, antigen assay, non-invasive

## Abstract

Severe acute respiratory syndrome coronavirus 2 (SARS-CoV-2) has caused considerable disruption worldwide. For efficient SARS-CoV-2 detection, new methods of rapid, non-invasive sampling are needed. This study aimed to investigate the stability of SARS-CoV-2 in a novel medium for gargle-lavage (GL) self-sampling and to compare the performance of SARS-CoV-2 detection in paired self-collected GL and clinician-obtained nasopharyngeal swab (NPS) samples. The stability study for SARS-CoV-2 preservation in a novel medium was performed over 14 days (4 °C, 24–27 °C, and 37 °C). In total, 494 paired GL and NPS samples were obtained at the University Hospital in Olomouc in April 2021. SARS-CoV-2 detection in paired samples was performed with a SARS-CoV-2 Nucleic Acid Detection Kit (Zybio, Chongqing Municipality, Chongqing, China), an Elecsys^®^ SARS-CoV-2 Antigen assay (Roche Diagnostics, Mannheim, Germany), and a SARS-CoV-2 Antigen ELISA (EUROIMMUN, Lübeck, Germany). The stability study demonstrated excellent SARS-CoV-2 preservation in the novel medium for 14 days. SARS-CoV-2 was detected in 55.7% of NPS samples and 55.7% of GL samples using rRT-PCR, with an overall agreement of 91.9%. The positive percent agreement (PPA) of the rRT-PCR in the GL samples was 92.7%, and the negative percent agreement (NPA) was 90.9%, compared with the NPS samples. The PPA of the rRT-PCR in the NPS and GL samples was 93.2% when all positive tests were used as the reference standard. Both antigen detection assays showed poor sensitivity compared to rRT-PCR (33.2% and 36.0%). rRT-PCR SARS-CoV-2 detection in self-collected GL samples had a similar PPA and NPA to that of NPSs. GL self-sampling offers a suitable and more comfortable alternative for SARS-CoV-2 detection.

## 1. Introduction 

Severe acute respiratory syndrome coronavirus 2 (SARS-CoV-2) causes coronavirus disease 2019 (COVID-19), a serious and potentially deadly disease [1]. Globally, there have been more than 640 million confirmed COVID-19 cases, including more than 6.61 million deaths reported to the WHO (https://covid19.who.int/, accessed on 6 December 2022). 

The standard format for SARS-CoV-2 detection is nasopharyngeal swab (NPS) sampling followed by real-time reverse transcription-polymerase chain reaction (rRT-PCR) [1,2]. However, NPS specimen collection requires trained medical personnel to be available and exposes them to a higher risk of transmission. Additionally, the nasopharyngeal swabbing method can cause discomfort to patients, especially those who need to be tested repeatedly or have an extremely sensitive mucous membrane, e.g., cancer patients [3,4]. Moreover, there are several contraindications, such as severe coagulopathy or nasal trauma [5]. Thus, a non-invasive sampling method that does not require medical intervention would greatly facilitate, speed up, and streamline the testing process. Several non-invasive methods have been validated. The most promising is gargle lavage sampling. This non-invasive and painless self-sampling method can be performed without contact with health care personnel. Several gargle lavage sampling techniques have been tested to date [6,7,8], but none of them have used a stabilization medium to preserve genomic and viral nucleic acid. In this study, a new collection medium was developed and validated for self-sampling and subsequent SARS-CoV-2 rRT-PCR testing. 

Moreover, gargle lavage samples obtained via this new sampling method were validated for antigen testing. Unlike PCR, antigen diagnostic tests are designed to directly detect SARS-CoV-2 capsid proteins. These tests are cheaper, faster, and do not require trained medical and laboratory personnel; however, they are significantly less sensitive than RT-PCR tests, especially for asymptomatic individuals [9,10]. The aim of the study was to validate a new gargling self-sampling device for SARS-CoV-2 detection via rRT-PCR and antigen assays. 

## 2. Materials and Methods

### 2.1. Stability of SARS-CoV-2 in GARGTEST Media 

To investigate the preservation of SARS-CoV-2 in a novel gargle lavage medium with a GARGTEST sampling kit (IntellMed Ltd., Olomouc, Czech Republic), we performed a stability study. The GARGTEST sampling tube contained lyophilised medium intended to be dissolved in 5 mL of gargle lavage (GL). 

The stability study was performed with SARS-CoV-2 spiked to both a simulated GL and GARGTEST medium dissolved in 5 mL of water. The simulated GL was prepared by pooling of three SARS-CoV-2 negative GL samples. Sampling was performed according to the manufacturer’s instructions (described below). The GARGTEST medium dissolved in water and the simulated GL sample was spiked with 62.23 PFU/mL SARS-CoV-2 at day 0. Stability was tested in three biological replicates of simulated samples stored at 4 °C, at room temperature (24–27 °C), and at 37 °C for 14 days (at day 0, 1, 4, 7, 11, and 14). Three biological replicates of the dissolved GARGTEST medium stored at 4 °C were used as the control.

For the nucleic acid extraction, a Viral Nucleic Acid Extraction Kit and an EXM6000 Zybio Nucleic Acid Isolation System (both Zybio Inc., Chongqing Municipality, Chongqing, China) were used according to the manufacturer’s instructions. A SARS-CoV-2 Nucleic Acid Detection Kit (PCR-Fluorescent Probe Method; Zybio Inc., Chongqing Municipality, Chongqing, China) was used for SARS-CoV-2 detection according to the manufacturer´s instructions (for further description see “rRT-PCR SARS-CoV-2 detection” below).

### 2.2. Clinical Study Design 

The study was conducted at the University Hospital, Olomouc, between 15 April 2021 and 28 April 2021. Patients who were sent for SARS-CoV-2 testing based on either symptoms or exposures were prospectively enrolled. Each study participant provided two types of samples: NPS and GL. All study participants and/or their legal guardians provided written informed consent. This study was performed in compliance with the Helsinki Declaration according to the study ethics proposal approved by the Ethics Committee of the Faculty of Medicine and Dentistry at Palacky University and the University Hospital in Olomouc (protocol no. 162/20). A flowchart of the study design is presented in Figure 1.

### 2.3. Sample Collection 

Four hundred and ninety-four paired NPS and GL samples were collected from patients aged from 3 to 76 years old (median 42). NPSs were collected by trained medical personnel using a flexible flocked swab and transported in viral transport media (ESwab collection system; Copan, Italy). After nasopharyngeal sampling, each participant performed self-sampling using GARGTEST (IntellMed Ltd., Olomouc, Czech Republic). The participants were provided with a container containing 5 mL of tap water, which they were asked to gargle for 20–30 s. After gargling, the sample was returned to the container, and then transferred to a tube containing the lyophilised medium and shaken vigorously. All NPS and GL samples were collected and analysed in parallel using rRT-PCR. Aliquots of the GL samples were heat-inactivated by incubation at 65 °C for 20 min and analysed by two different antigen detection tests. If a conclusive result was obtained from the test, clinical samples were tested using rRT-PCR as well as antigen detection tests in one replicate.

### 2.4. rRT-PCR SARS-CoV-2 Detection 

Nucleic acid extraction was performed using a Viral Nucleic Acid Extraction Kit (Zybio Inc., Chongqing, China) and an EXM6000 Zybio Nucleic Acid Isolation System according to the manufacturer’s recommendations. RNA extracted from 200 µL of the primary sample was eluted in 50 µL of elution buffer. SARS-CoV-2 was detected using a SARS-CoV-2 Nucleic Acid Detection Kit (PCR-Fluorescent Probe Method; Zybio Inc., Chongqing, China) according to the manufacturer’s recommendations. This kit uses SARS-CoV-2 specific primers and probes to detect the *E* gene (gene for envelope protein), *N* gene (gene for nucleocapsid protein) and *RdRP* gene (gene for RNA-dependent RNA polymerase) of SARS-CoV-2, and primers and probe to detect the human *GAPDH* gene (gene for human glyceraldehyde-3-phosphate dehydrogenase) as an internal control (IC) to monitor the whole process, including sampling. Criteria for the evaluation of the results are described in Table 1.

### 2.5. Antigen-Detection Diagnostic Tests 

First, 1 mL aliquots of GL samples were heat-inactivated by incubation at 65 °C for 20 min. Heat-inactivated samples were examined using an Elecsys^®^ SARS-CoV-2 Antigen assay (Roche Diagnostics, Mannheim, Germany; herein referred to as “Elecsys^®^”) according to the manufacturer’s recommendations. Samples were evaluated as SARS-CoV-2 positive (reactive) if the cut-off-index (COI) was ≥1. Samples that failed to be analysed were re-tested. A SARS-CoV-2 Antigen ELISA (EUROIMMUN, Lübeck, Germany; herein referred to as “SARS-CoV-2 Antigen ELISA”) assay was also used to test all heat-inactivated GL samples according to the manufacturer’s instructions. Samples were evaluated as SARS-CoV-2-positive if the sample to calibration extinction ratio was ≥0.50. The manufacturer’s specifications classified borderline positive samples as those with a sample to calibration extinction ratio of 0.50 ≥ *N* < 0.60, and positive samples as those with a sample to calibration extinction ratio of ≥0.60.

### 2.6. Statistical Analysis 

The statistical software R (version 4.1.0; R Core Team, R Foundation for Statistical Computing [http://www.r-project.org], accessed on 9 June 2021) was used for data evaluation. Positive percentage agreement (PPA), negative percentage agreement (NPA), and overall agreement (OA) were used for estimating an agreement of test results to a non-reference standard (NPS rRT-PCR, GL sample rRT-PCR mutually, resp. to combined results) with 95% confidence limits calculated by the Clopper–Pearson method (GenBinomApps R package, ver. 1.1).

Sensitivity, specificity, predictive values, and κ coefficients (epiR, ver 2.0.19 R Package) were calculated using the GL sample rRT-PCR results as a reference for GL sample antigen testing. To improve the diagnostic power of the antigen assays for GL samples, a new cut-off was estimated by the optimal.cutpoints function (OptimalCutpoints, ver. 1.1-4 R Package) and maxSpSe method (maximising sensitivity and specificity simultaneously). The Elecsys^®^ SARS-CoV-2 Antigen and SARS-CoV-2 Antigen ELISA assays’ relative sensitivity for SARS-CoV-2 detection in positive GL samples using the original and optimised cut-off values for the subsets, according to the disjoint intervals of the Ct *N* gene values, were estimated with the prop.test function without Yates’ continuity correction. The Student’s *t*-test and Wilcoxon test were used to compare of the distribution of Ct values of the examined genes between the groups of concordant and discordant GL samples, respectively. NPS sample *p*-values were adjusted (across genes) with the Bonferroni method.

## 3. Results

### 3.1. Stability of SARS-CoV-2 in GARGTEST Gargle Lavage Media 

We found that SARS-CoV-2 had good stability in all the tested samples. Simulated GL samples were extremely stable at 4 °C, with no change in Ct values after 14 days. Additionally, at room temperature (24–27 °C) and at 37 °C, the stability of SARS-CoV-2 after 14 days was very good with Ct value average differences of 4.08 and 5.75 (Figure 2, Appendix A).

### 3.2. Comparison of SARS-CoV-2 Detection in NPS and GL Samples Using rRT-PCR 

Paired NPS and GL samples were obtained from 494 patients. SARS-CoV-2 was detected in 55.7% of NPS samples (275 of 494). Similarly, 55.7% (275 of 494) of GL samples were SARS-CoV-2-positive (Table 2). Out of 494 samples, 15 NPS (3.04%) and 12 GL samples (2.43%) gave inconclusive results due to positivity for only one or two SARS-CoV-2 target genes and had to be re-extracted and re-tested. After repeated testing, 13 NPS samples were classed as SARS-CoV-2-positive and 2 as SARS-CoV-2-negative. Of the 12 GL samples initially determined inconclusive, 11 were classed as SARS-CoV-2 positive and 1 was classed as SARS-CoV-2 negative.

In 8.1% of cases (40 of 494), only one of the paired samples was SARS-CoV-2 positive. Twenty patients were SARS-CoV-2-positive in NPS samples only, and another twenty patients were classed as SARS-CoV-2-positive in GL samples only (Table 2). The median Ct value for all detected SARS-CoV-2 genes was higher in samples discordant with paired samples than in samples concordant with paired samples for both NPS and GL (Table 3 and Figure 3).

The median NPS Ct value for *GAPDH* in NPSs negative for SARS-CoV-2 but positive in the paired GL samples was higher than in the concordant NPS samples (29.4 vs. 27.12, *p* = 0.011). The median GL Ct value for *GAPDH* in the GL samples negative for SARS-CoV-2 but positive in the paired NPS samples was not significantly different from the GL samples with a concordant GL and NPS result (24.89 vs. 24.11, *p* = 0.948) (Table 3 and Figure 4).

There was good agreement for SARS-CoV-2 detection between the self-sampled GL samples and the conventional NPS samples with an overall accuracy of 91.9% (89.1%, 94.2%). When rRT-PCR SARS-CoV-2 detection in the NPS samples was considered as a non-reference standard, the positive percent agreement (PPA) for SARS-CoV-2 detection in GL samples was 92.7% (95% CI: 89.1% to 94.2%) and the negative percent agreement (NPA) was 90.9% (95% CI: 86.2% to 94.3%). The PPA and NPA value in the NPS and GL samples were 93.2% (95% CI: 89.7% to 95.8%) when all positive tests were used as the reference standard (Table 4).

### 3.3. Comparison of SARS-CoV-2 Detection in GL Samples Using rRT-PCR and Antigen-Detection Diagnostic Tests 

In parallel, 494 GL samples were tested with rRT-PCR, the Elecsys^®^ SARS-CoV-2 Antigen assay, and the SARS-CoV-2 Antigen ELISA assay. All (209/209) rRT-PCR-negative GL samples gave negative results using the Elecsys^®^ assay (Table 5). Only 33.2% of the GL samples (89/268) classed as positive by rRT-PCR were found to be positive by Elecsys^®^; the remaining 179 rRT-PCR-positive samples were negative using Elecsys^®^. Seventeen samples failed to be analysed by Elecsys^®^ because of the high viscosity of the samples.

Compared with rRT-PCR SARS-CoV-2 detection, Elecsys^®^ reached 100% specificity; however, its sensitivity was only 33.2%. There was poor agreement for SARS-CoV-2 detection between rRT-PCR and Elecsys^®^ (κ = 30.3%; 95% CI, 23.9% to 36.8%) with 62.5% accuracy. The highest sensitivity and specificity were observed when only samples with Ct ≤ 25 were evaluated (95.7% and 89.8%, respectively). Values of sensitivity and specificity for various groups of samples are summarised in Table 5 and Appendix A. The maximum sensitivity and specificity for the unselected samples were observed when an optimised cut-off value was used. When COI = 0.675 was used as the cut-off for SARS-CoV-2 positivity in Elecsys^®^, the sensitivity was 70.1% and the specificity was 70.8% (Table 6, Appendix A). Values of sensitivity and specificity using COI = 0.675 for various groups of samples are summarised in Table 6 and Appendix A.

Compared with rRT-PCR SARS-CoV-2 detection, the SARS-CoV-2 Antigen ELISA reached 36.0% sensitivity and 95.4% specificity. There was a poor agreement for SARS-CoV-2 detection between rRT-PCR and the SARS-CoV-2 Antigen ELISA (κ = 29.2%; 95% CI, 22.3% to 36.0%), with 62.3% accuracy. The highest sensitivity and specificity were observed when only samples with Ct ≤ 25 were evaluated (89.8% and 85.4%, respectively). Values of sensitivity and specificity for various groups of samples are summarised in Table 7 and Appendix A. The maximum sensitivity and specificity for unselected samples was observed when an optimised cut-off value was used. When a sample with a calibration extinction ratio of 0.4 was used as a cut-off for SARS-CoV-2 positivity in the SARS-CoV-2 Antigen ELISA, the sensitivity was 62.2% and the specificity was 65.3% (Table 8, Appendix A). Sensitivity and specificity values using a sample with a calibration extinction ratio of 0.4 for various groups of samples are summarised in Table 8 and Appendix A.

## 4. Discussion

Precise diagnostic methods and the acquisition of optimal clinical specimens for the detection of SARS-CoV-2 are crucial for containing the COVID-19 pandemic [2,11]. However, collecting NPS samples is an unpleasant, time- and resource-consuming process requiring trained healthcare workers. A few recent studies have examined alternative sampling methods, such as saliva [12,13] and saline gargle samples [6,7,14], sputum, urine and stool samples [15,16]. Our study evaluated self-collected GL samples using a new GARGTEST transport medium as an alternative sampling method to conventional NPSs for SARS-CoV-2 rRT-PCR. This study also aimed to evaluate SARS-CoV-2 detection by antigen detection with GL, since there is very limited knowledge on the combination of GL and antigen assays.

In our study, the rRT-PCR detection of SARS-CoV-2 in self-collected GL samples with NPS samples as a non-reference standard showed a high PPA, NPA, and good overall agreement between conventional NPS and GL samples. However, the comparison to NPS may not be optimal because the virus is detectable only in the oral cavity. Recent results suggest that oral area sampling could be more effective than NPS in omicron variant testing [17]. Therefore, we also compared the results of GL and NPS samples to the reference which was SARS-CoV-2 positivity in at least one of the sample types. The PPA of both GL and NPS samples was higher. SARS-CoV-2 detection in GL samples was therefore found to be as reliable as in NPSs. To date, only several studies have compared SARS-CoV-2 detection in self-collected oral gargle samples and NPSs [6,7,8,18,19,20,21,22].

In a small Canadian study enrolling 40 SARS-CoV-2-positive individuals, paired NPS, saline gargle samples and saliva samples were compared. Gargle samples showed >97% sensitivity, whereas saliva samples showed a sensitivity of 79%. Moreover, oral gargle samples were more acceptable to patients than NPS and saliva collection [6]. Another study comparing NPSs taken by trained specialists and self-collected GL (*n* = 80) reported 100% congruence [7]. All 26/80 SARS-CoV-2-positive individuals tested positive in both paired samples. GL and NPS sampling was also shown to produce similar amounts of the primary sample. In line with the results of this study, we confirmed that the amount of biological material in oral/throat GL is sufficient for SARS-CoV-2 diagnostic tests despite the high dilution rate. Based on Ct values of housekeeping gene detection, RNA levels were even slightly higher in oral gargle samples than in NPSs. Moreover, while no difference was observed in GL Ct values of the *GAPDH* gene in concordant and discordant samples, a significant difference was found for NPS Ct values between concordant and discordant cases, indicating poor NPS sampling (Figure 4).

In Kohmer et al. ’s 2021 study, five different self-sample types were compared to NPS using a set of 102 individual samples [8]. The GL using tap water was the second most sensitive (89.1%) sampling method after the saliva collection. The lower sensitivity of GL in this study could be caused by the delay between NPS and GL collection (up to 48 h). In our study, NPS and GL were collected at the same time.

The only study recruiting a similar number of outpatients (*n* = 608) to the present study reported a slightly lower sensitivity (89%) but higher specificity (> 99%). In contrast to our study, the SARS-CoV-2 positivity rate was only 9.4%. Nevertheless, similarly to our study, the median age of participants with COVID-19 was 33 years, and children <18 years were enrolled [19].

Oral gargle sampling is a time-saving and easy technique, with minimal requirements for the assistance of trained specialists. Self-collected oral gargle samples could be an effective tool for preventing SARS-CoV-2 transmission, even among children under 18 years of age at school, or untrained individuals at mass social events [6,19,23]. However, for SARS-CoV-2 detection, oral rinse samples should be distinguished from oral gargle samples. A recent study showed that oral rinse samples had a significantly lower sensitivity (63.6%) and specificity (96.9%) for SARS-CoV-2 detection [24] than reported in our and other studies [6,7,19].

The specimen stability at room temperature is another advantage of self-collected gargle samples over conventional NPSs. It has been reported that SARS-CoV-2 RNA in saline oral gargle samples is stable at room temperature for at least 2 days from specimen acquisition [6]. The stability study of GARGTEST medium used in our study showed the stability of SARS-CoV-2 RNA for at least 14 days at 4–37 °C. Prolonged SARS-CoV-2 stability for up to 31 days in commercially available sampling media was recently reported [22], however, such a long stability is not very useful in clinical practice.

Although GL samples were evaluated for use in the preceding SARS (SARS-CoV-1) pandemic in 2002–2004 [25,26], only one study evaluating self-collected GL samples for SARS-CoV-2 detection by any antigen assay has been published. Kheiroddin et al. [27] reported reliable SARS-CoV-2 detection with a STANDARD™ F COVID-19 Ag FIA kit (SD BIOSENSOR Inc., Suwon, Korea) for GL samples with Ct < 20. Antigen testing in samples with Ct values above this limit showed poor results or failed. Until today, no other study evaluating antigen-based SARS-CoV-2 detection in GL has been published. However, the Elecsys^®^ SARS-CoV-2 Antigen assay has already been evaluated for saliva samples. They reached higher sensitivity in samples with Ct ≤ 30 than we did (78.6% vs. 54.7%), but the sensitivity in samples with Ct ≤ 26 was comparable to our results (100% vs. 95.7%) when non-optimised COI was used [28]. This sensitivity drop is likely caused by the higher dilution of GL samples compared to saliva.

In general, a lower performance of COVID-19 antigen assays is reported even for assays combined with NPSs compared with rRT-PCR methods. Most antigen assays are designed to be used in combination with NPSs, alternatively, oropharyngeal swabs or saliva samples [29,30]. The combination of easily self-collected gargle samples and rapid SARS-CoV-2 antigen detection assays could enable improvements and new opportunities for containing the COVID-19 pandemic. Nevertheless, no antigen assay for SARS-CoV-2 antigen detection with GL are available. Although GL samples are comparable to NPSs for SARS-CoV-2 detection using rRT-PCR methods, they are too diluted for the antigen-based detection of SARS-CoV-2, which is less sensitive than rRT-PCR. As we show in our study, neither the Elecsys^®^ assay nor SARS-CoV-2 Antigen ELISA assay meet the minimum performance requirements for antigen-detection assays specified by the WHO [31]. Thus, further research is needed, particularly as the overall sensitivity and specificity of antigen tests for saliva samples are highly variable and complex, depending on sample collection, preparation, and the type of antigen assay chosen for SARS-CoV-2 detection [32,33,34,35]. For more reliable SARS-CoV-2 detection with GL, the design of specific antigen assays with optimised COI for GL samples would be beneficial.

Our study design has several strengths. First, as well as the high number of paired samples (*n* = 494), all individuals enrolled in this study were outpatients, including symptomatic and asymptomatic patients, and of all age cohorts (>3 years). Second, all paired NPS and GL samples were tested in parallel using the same isolation and rRT-PCR platform within 24 h of acquisition, thereby eliminating inaccuracy as result of variable analytic thresholds. Moreover, all the GL samples were collected using a novel validated medium, which ensured long-term sample stability at room temperature. Beyond the COVID-19 pandemic, GARGTEST sampling medium may be useful for the detection of other pathogens or biomarkers from the upper aerodigestive tract. Currently, this medium is under evaluation for other oral microbial infections. The main limitation of this study was that some patients recruited in our study did not have samples collected within the first 5 days of illness.

In conclusion, we demonstrated that self-collected oral gargle lavage samples are a reliable alternative to nasopharyngeal swabs for SARS-CoV-2 detection by rRT-PCR. Gargle lavage sampling eliminates most of the inconveniences of NPSs and offers improved efficiency for managing the COVID-19 pandemic. The novel sampling medium may be useful for the detection of other oral cavity pathogens and biomarker monitoring which is the subject of ongoing research.

## Figures and Tables

**Figure 1 viruses-14-02829-f001:**
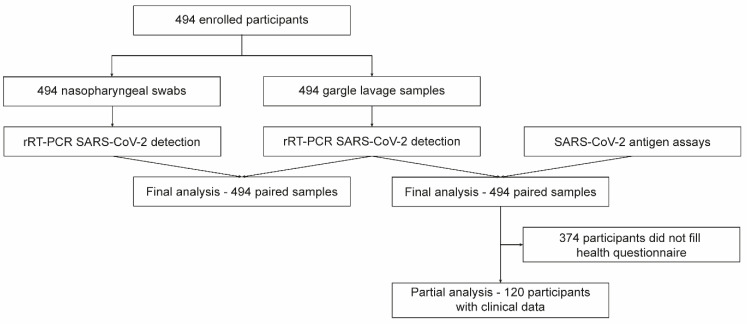
Flowchart of the study.

**Figure 2 viruses-14-02829-f002:**
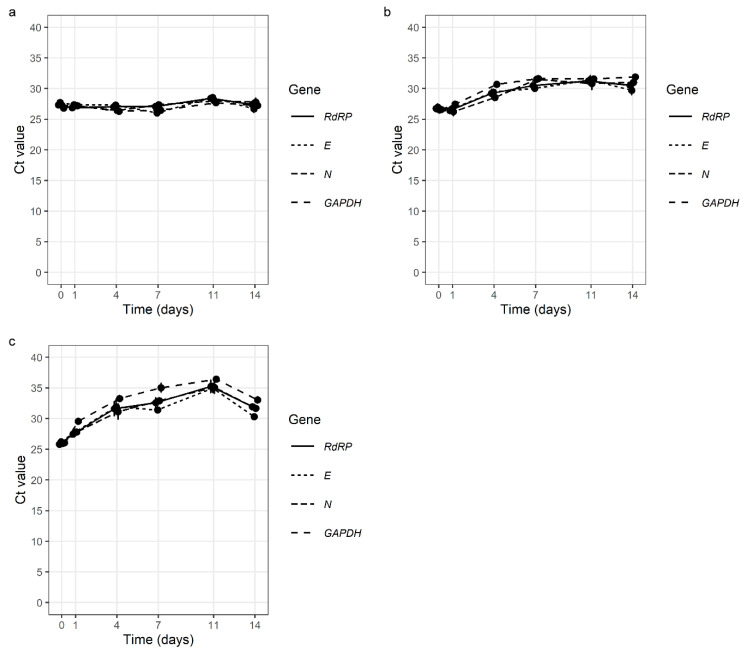
Stability of SARS-CoV-2 in simulated gargle lavage samples at 4 °C (**a**), at room temperature (24 °C–27 °C) (**b**), and at 37 °C (**c**) over 14 days (*E* gene—*E* gene of SARS-CoV-2; *N* gene—*N* gene of SARS-CoV-2 *RdRP*—gene for RNA-dependent RNA polymerase of SARS-CoV-2; *GAPDH*—gene for human glyceraldehyde-3-phosphate dehydrogenase (internal control).

**Figure 3 viruses-14-02829-f003:**
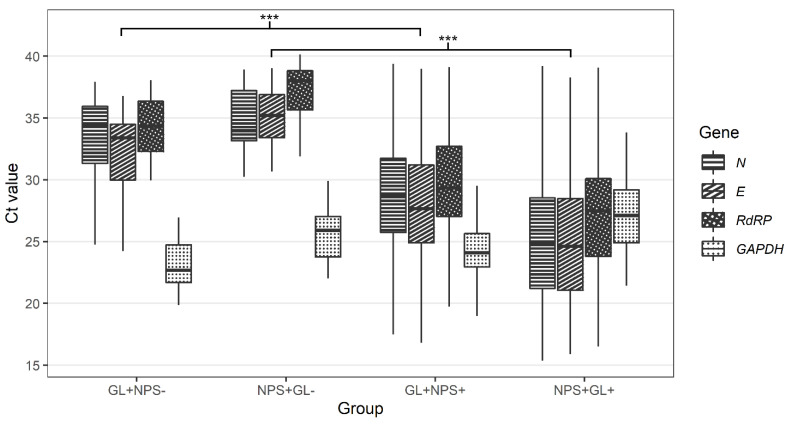
Comparison of Ct values for *N*, *E*, *RdRP,* and *GAPDH* genes in nasopharyngeal and gargle lavage samples with concordant and discordant results (GL+NPS−-—Ct values for gargle lavage samples for cases found positive only with gargle lavage samples; NPS+GL−-Ct values for nasopharyngeal swabs for cases found positive only with nasopharyngeal swabs; GL+NPS+-Ct values for gargle lavage samples for cases found positive with both gargle and nasopharyngeal swabs; NPS+GL+-Ct values for nasopharyngeal swabs for cases found positive with both gargle and nasopharyngeal swabs; *E* gene-*E* gene of SARS-CoV-2; *N* gene-*N* gene of SARS-CoV-2; *RdRP*-gene for RNA-dependent RNA polymerase of SARS-CoV-2; *GAPDH*-gene for human glyceraldehyde-3-phosphate dehydrogenase (internal control). Calculated by Wilcoxon two-sample test (GL+NPS+ vs. GL+NPS-, NPS+GL+ vs. NPS+GL-) with Bonferroni correction (across genes). *** *p* < 0.001.

**Figure 4 viruses-14-02829-f004:**
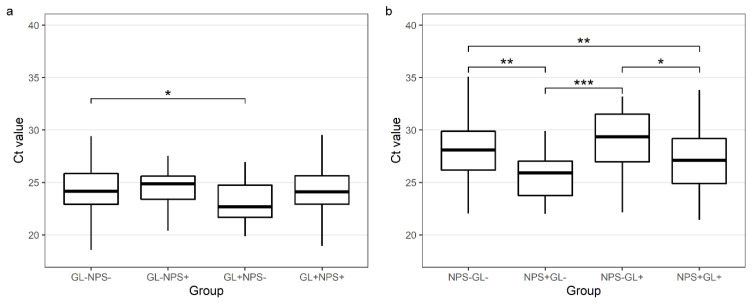
Comparison of Ct values for *GAPDH* genes in gargle lavage samples (**a**) and nasopharyngeal swabs (**b**) with concordant and discordant results (**a**) GL−NPS−-*GAPDH* Ct values for gargle lavage samples for cases found negative with both gargle and nasopharyngeal swabs; GL−NPS+-*GAPDH* Ct values for gargle lavage samples for cases found positive only with nasopharyngeal swabs; GL+NPS−-*GAPDH* Ct values for gargle lavage samples for cases found positive only with gargle lavage samples; GL+NPS+-*GAPDH* Ct values for gargle lavage samples for cases found positive with both gargle and nasopharyngeal swabs. (**b**) NPS−GL−-*GAPDH* Ct values for nasopharyngeal swabs for cases found negative with both gargle and nasopharyngeal swabs; NPS+GL−-*GAPDH* Ct values for nasopharyngeal swabs for cases found positive only with nasopharyngeal swabs; NPS−GL+-*GAPDH* Ct values for nasopharyngeal swabs for cases found positive only with gargle lavage samples; NPS+GL+-*GAPDH* Ct values for nasopharyngeal swabs for cases found positive with both gargle and nasopharyngeal swabs. Calculated by the Tukey test (after ANOVA). * *p* < 0.05, ** *p* < 0.01, *** *p* < 0.001.

**Table 1 viruses-14-02829-t001:** Criteria used for the evaluation of rRT-PCR (SARS-CoV-2 nucleic acid detection kit) results.

Result	Criteria
SARS-CoV-2 positive	All three SARS-CoV-2 target genes detected (Ct ≤ 41).
SARS-CoV-2 negative	No SARS-CoV-2 target gene detected (Ct > 41), Ct for IC < 40.
	Only one target gene detected (Ct > 38), Ct for IC < 40.
Inconclusive result	Two target genes detected (Ct ≤ 41) or one target gene detected (Ct ≤ 38): - Repeat test including RNA extraction. - When at least one target gene is detected (Ct ≤ 41), sample is evaluated as SARS-CoV-2-positive.
	Ct for IC > 40: - Repeat test including RNA extraction.
Poor sampling	Even after repeated testing Ct for IC > 40: - New sampling recommended.

IC—internal control.

**Table 2 viruses-14-02829-t002:** Results of rRT-PCR SARS-CoV-2 detection in paired nasopharyngeal swab and gargle lavage samples.

		Nasopharyngeal Swab (NPS)		
		Positive	Negative	Total
Gargle lavage (GL)	Positive	255 (51.6%)	20 (4.05%)	275 (55.7%)
	Negative	20 (4.05%)	199 (40.3%)	219 (44.3%)
	Total	275 (55.7%)	219 (44.3%)	494

**Table 3 viruses-14-02829-t003:** Comparison of median Ct values in nasopharyngeal swab and gargle lavage samples based on the agreement of SARS-CoV-2 test results with paired samples.

Detected Gene	Nasopharyngeal Swab						Gargle Lavage					
	Concordant		Discordant		*p*-Value *	*p*-Value ^§^	Concordant		Discordant		*p*-Value *	*p*-Value ^§^
	NPS+GL+ *^§^	NPS−GL−	NPS+GL−	NPS−GL+ ^§^			GL+NPS+*^§^	GL−NPS−	GL+NPS−	GL−NPS+ ^§^		
*E* gene	24.63	NA	35.21	NA	<0.001 *	NA	27.67	NA	33.40	NA	<0.001 *	NA
*N* gene	24.84	NA	33.98	NA	<0.001 *	NA	28.69	NA	34.44	NA	<0.001 *	NA
*RdRP* gene	27.49	NA	38.03	NA	<0.001 *	NA	29.35	NA	34.35	NA	<0.001 *	NA
*GAPDH*	27.12	28.11	25.92	29.36	0.105 *	0.011	24.11	24.16	22.69	24.89	0.112 *	0.948

*E* gene-*E* gene of SARS-CoV-2; *N* gene-*N* gene of SARS-CoV-2; *RdRP*-gene for RNA-dependent RNA polymerase of SARS-CoV-2; *GAPDH*-gene for human glyceraldehyde-3-phosphate dehydrogenase; GL-gargle lavage; NPS-nasopharyngeal swab; NA-not applicable; NPS+GL+—-Ct values for nasopharyngeal swabs for cases found positive with both gargle and nasopharyngeal swabs; NPS+GL−-Ct values for nasopharyngeal swabs for cases positive only with nasopharyngeal swabs; GL+NPS—-Ct values for gargle lavage samples for cases found positive only with gargle lavage samples; GL+NPS+-Ct values for gargle lavage samples for cases found positive with both gargle and nasopharyngeal swabs. * *p*-value for comparison of NPS+GL+ vs. NPS+GL− and GL+NPS+ vs. GL+NPS− Ct values of all tested genes. Calculated using the McNemar test. ^§^
*p*-value for comparison of NPS−GL+ vs. NPS+GL+ and GL−NPS+ vs. GL+NPS+ *GAPDH* Ct values. Calculated using the Tukey test.

**Table 4 viruses-14-02829-t004:** Performance characteristics of SARS-CoV-2 rRT-PCR detection in nasopharyngeal swab and gargle lavage samples using different reference standards.

Sample Type	Reference Standard	PPA (95% CI)	NPA (95% CI)	OA (95% CI)
GL	NPS	92.7% (89.0%, 95.5%)	90.9% (86.2%, 94.3%)	91.9% (89.1%, 94.2%)
GL	NPS+GL	93.2% (89.7%, 95.8%)	100% (98.5%, 100%)	96.0% (93.8%, 97.5%)
NPS	GL	92.7% (89.0%, 95.5%)	90.9% (86.2%, 94.3%)	91.9% (89.1%, 94.2%)
NPS	NPS+GL	93.2% (89.7%, 95.8%)	100% (98.5%, 100%)	96.0% (93.8%, 97.5%)

GL-gargle lavage; NPS-nasopharyngeal swab PPA-positive percent agreement; NPA-negative percent agreement; OA-overall agreement, CI-confidence interval; NPS+GL-NPS and/or GL tested positive.

**Table 5 viruses-14-02829-t005:** Results of reference method (rRT-PCR) and tested method (Elecsys^®^ SARS-CoV-2 Antigen assay) using the manufacturer’s specified cut-off value.

Elecsys^®^ (Original COI)	rRT-PCR			rRT-PCR (*N* Gene Ct ≤ 25 *)			rRT-PCR (*N* Gene Ct ≤ 30 ^#^)			rRT-PCR (Symptomatic Patients ^§^)		
	Pos.	Neg.	Total	Pos.	Neg.	Total	Pos.	Neg.	Total	Pos.	Neg.	Total
Pos. (COI ≥ 1)	89	0	89	45	44	89	88	1	89	42	0	42
Neg. (COI < 1)	179	209	388	2	386	388	73	315	388	80	30	110
Total	268	209	477 †	47	430	477 †	161	316	477 †	122	30	152
Sensitivity	33.2%			95.7%			54.7%			34.4%		
Specificity	100.0%			89.8%			99.7%			100.0%		
NPV	53.9%			99.5%			81.2%			27.3%		
PPV	100.0%			50.6%			98.9%			100%		
Accuracy	62.5%			90.4%			84.5%			47.4%		
ĸ (95% CI)	30.3% (23.9%–36.8%)			61.2% (52.8%–69.6%)			61.0% (52.7%–69.3%)			17.2% (8.3%–26.1%)		
*p*-value ^‡^	<0.001			<0.001			<0.001			<0.001		

COI—cut-off index; Pos.—positive; PPV—positive predictive value; Neg.—negative; NPV—negative predictive value. * Only samples with Ct ≤ 25 for *N* gene were evaluated as positive. ^#^ Only samples with Ct ≤ 30 for *N* gene were evaluated as positive. ^§^ Only patients within the first 5 days of illness included in the analysis. † Seventeen samples failed to by analysed by Elecsys^®^ because of the viscosity of the sample. ^‡^ *p*-value calculated using the McNemar test.

**Table 6 viruses-14-02829-t006:** Results of reference method (rRT-PCR) and tested method (Elecsys^®^ SARS-CoV-2 Antigen assay) using an optimised cut-off value.

Elecsys^®^ (Optimised COI)	rRT-PCR			rRT-PCR (*N* Gene Ct ≤ 25 *)			rRT-PCR (*N* Gene Ct ≤ 30 ^#^ )			rRT-PCR (Symptomatic Patients ^§^)		
	Pos.	Neg.	Total	Pos.	Neg.	Total	Pos.	Neg.	Total	Pos.	Neg.	Total
Pos. (COI ≥ 0.675)	188	61	249	47	202	249	148	101	249	88	14	102
Neg. (COI < 0.675)	80	148	228	0	228	228	13	215	228	34	16	50
Total	268	209	477 ^†^	47	430	477 ^†^	161	316	477 †	122	30	152
Sensitivity	70.1%			100.0%			91.9%			72.1%		
Specificity	70.8%			53.0%			68.0%			53.3%		
NPV	64.9%			100.0%			94.3%			32.0%		
PPV	75.5%			18.9%			59.4%			86.3%		
Accuracy	70.4%			57.7%			76.1%			68.4%		
ĸ (95% CI)	40.6% (31.6%–49.5%)			18.2% (13.0%–23.4%)			52.9% (44.5%–61.2%)			20.3% (5.4%–35.3%)		
*p*-value ^‡^	0.13			<0.001			<0.001			0.006		

COI—cut-off index; Pos.—positive; PPV—positive predictive value; Neg.—negative; NPV—negative predictive value. * Only samples with Ct ≤ 25 for *N* gene were evaluated as positive. ^#^ Only samples with Ct ≤ 30 for *N* gene were evaluated as positive. ^§^ Only patients within the first 5 days of illness included in the analysis. ^†^ Seventeen samples failed to by analysed by Elecsys^®^ because of the viscosity of the samples. ^‡^ *p*-value calculated using the McNemar test. Out of 494 tested samples, 209/219 (95.4%) rRT-PCR-negative GL samples gave negative results using the SARS-CoV-2 Antigen ELISA assay, and 10 rRT-PCR negative samples were classed as positive by the SARS-CoV-2 Antigen ELISA (Table 7). Only 99/275 (36.0%) GL samples found positive by rRT-PCR were found to be positive by the SARS-CoV-2 Antigen ELISA; the remaining 176 rRT-PCR-positive samples were negative using the SARS-CoV-2 Antigen ELISA.

**Table 7 viruses-14-02829-t007:** Results of reference method (rRT-PCR) and tested method (SARS-CoV-2 Antigen ELISA) using the manufacturer’s specified cut-off value.

SARS-CoV-2 Antigen ELISA (Original COI)	rRT-PCR			rRT-PCR (Ct ≤ 25 for *N* Gene *)			rRT-PCR (*N* Gene Ct ≤ 30 ^#^)			rRT-PCR (Symptomatic Patients ^§^)		
	Pos.	Neg.	Total	Pos.	Neg.	Total	Pos.	Neg.	Total	Pos.	Neg.	Total
Pos. (≥0.5)	99	10	109	44	65	109	90	19	109	52	0	52
Neg. (<0.5)	176	209	385	5	380	385	75	310	385	77	31	108
Total	275	219	494	49	445	494	165	329	494	129	31	160
Sensitivity	36.0%			89.8%			54.5%			40.3%		
Specificity	95.4%			85.4%			94.2%			100%		
NPV	54.3%			98.7%			80.5%			28.7%		
PPV	90.8%			40.4%			82.6%			100%		
Accuracy	62.3%			85.8%			81.0%			51.9%		
ĸ (95% CI)	29.2% (22.3%–36.0%)			48.7% (40.8%–56.6%)			53.3% (44.8%–61.7%)			20.7% (11.3%–30.2%)		
*p*-value ^†^	<0.001			<0.001			<0.001			<0.001		

COI—cut-off index; Pos.—positive; PPV—positive predictive value; Neg.—negative; NPV—negative predictive value. * Only samples with Ct ≤ 25 for *N* gene were evaluated as positive. ^#^ Only samples with Ct ≤ 30 for *N* gene were evaluated as positive. ^§^ Only patients within the first 5 days of illness included in the analysis. ^†^ *p*-value calculated using the McNemar test.

**Table 8 viruses-14-02829-t008:** Results of reference method (rRT-PCR) and tested method (SARS-CoV-2 Antigen ELISA) using an optimised cut-off value.

SARS-CoV-2 Antigen ELISA (Optimised COI)	rRT-PCR			rRT-PCR (*N* Gene Ct ≤ 25 *)			rRT-PCR (*N* Gene Ct ≤ 30 ^#^ )			rRT-PCR (Symptomatic Patients ^§^)		
	Pos.	Neg.	Total	Pos.	Neg.	Total	Pos.	Neg.	Total	Pos.	Neg.	Total
Pos. (≥0.4)	171	76	247	47	200	247	122	125	247	87	7	94
Neg. (<0.4)	104	143	247	2	245	247	43	204	247	42	24	66
Total	275	219	494	49	445	494	165	329	494	129	31	160
Sensitivity	62.2%			95.9%			73.9%			67.4%		
Specificity	65.3%			55.1%			62.0%			77.4%		
NPV	57.9%			99.2%			82.6%			36.4%		
PPV	69.2%			19.0%			49.4%			92.6%		
Accuracy	63.6%			59.1%			66.0%			69.4%		
ĸ (95% CI)	27.1% (18.4%–35.9%)			18.2% (12.9%–23.5%)			32.0% (23.7%–40.3%)			31.4% (17.9%–44.9%)		
*p*-value ^†^	0.044			<0.001			<0.001			<0.001		

COI—cut-off index; Pos.—positive; PPV—positive predictive value; Neg.—negative; NPV—negative predictive value. * Only samples with Ct ≤ 25 for *N* gene were evaluated as positive. ^#^ Only samples with Ct ≤ 30 for *N* gene were evaluated as positive. ^§^ Only patients within the first 5 days of illness included in the analysis. ^†^ *p*-value calculated using the McNemar test.

## Data Availability

The datasets generated during and/or analysed during the current study are available from the corresponding author on reasonable request.

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
