# Peer review of "Evaluation of Non-Invasive Gargle Lavage Sampling for the Detection of SARS-CoV-2 Using rRT-PCR or Antigen Assay"

_viruses, 2022, doi:10.3390/v14122829_

Round 1

Reviewer 1 Report

The manuscript entitled "Evaluation of non-invasive gargle lavage sampling for the detection of SARS-CoV-2 using rRT-PCR or antigen assay" describes the use and benefit of self-sampling. 

The manuscript is interesting and contains useful information. But I believe the manuscript required major revision before publication.

The introduction and conclusion parts are underdeveloped.

The authors should discuss the previous reports and compare or justify their work. How this is better or more beneficial?

Important references are included

for examples

Scientific Reports | (2022) 12:3706

J. Clin. Med. 202110(24), 5751

The authors should do exhaustive literature and provide a rationale for their work with a strong justified conclusion.

Author Response

  1. The introduction and conclusion parts are underdeveloped.

Response: Requirements have been addressed. We improved both parts of text. Thank you for the recommendation.

  1. The authors should discuss the previous reports and compare or justify their work. How this is better or more beneficial?

Important references are included

for examples

Scientific Reports | (2022) 12:3706

gJ. Clin. Med. 2021, 10(24), 5751

Response: Requirements have been addressed. We added more recent references. The Kohmer et al., 2021 have been already used as citation 8.

  1. The authors should do exhaustive literature and provide a rationale for their work with a strong justified conclusion.

Response: Requirements have been addressed. We added more recent references and improved the conclusion.

Reviewer 2 Report

The authors performed an interesting comparison of a large group of paired nasopharyngeal swabs and self-collected gargle-lavage (GL) for the detection of SARS-CoV-2 RNA, evidencing a high concordance between the two different types of specimens. The results are clear and reported in detail. 

I have some points that the authors could improve:

I will move Fig. 5 and 6 in Supplements

Results from line 151 to 160 and from line 169 to 172 should be summarized in Tables. 

Table 2 could be also improved

In Table 4,5,6, and 7 the authors could try to short the titles, for example: sample with Ct <= 25 for N gene evaluated as positive could be changed in N gene Ct <=25 und also positive, negative, and total in pos, neg, tot. Some information can be eventually moved in the legend. 

The discussion is too long and not always well written, it has to be revised. Some parts are just a repetition of results, these parts could be removed. In general, it should be more stringent. The first lines 246 to 255 should be shortened. The same for lines 291-296.

Line 269 the sentence is not clear

Line 271 is sufficient to??

Line 272 this sentence is not necessary ((GAPD) an an internal control Figure 3) could be changed with housekeeping gene detection, RNA levels ..

Line 276 "in other study" should be revised 

Reviewer 3 Report

Overall, the manuscript is promising however it needs improvement prior to recommendation. 

Comments: 

1. What is novel in your manuscript? Since early 2021, there have been studies already on samples for SARS-CoV-2 detection collected through gargling. Add this to your discussion.

2. Provide statistical significance to your figures. 

3. Include the statistical significance of the diagnostic tests conducted. Also, add the number of replicates for each test or the number of times it was tested. 

4. Discuss the reasons why the sensitivity of the test is very low? What contributes to this occurrence? 

Author Response

  1. What is novel in your manuscript? Since early 2021, there have been studies already on samples for SARS-CoV-2 detection collected through gargling. Add this to your discussion.

Response: The requirement has been addressed. We added more recent references to the discussion part and we also improved the introduction.

“Several non-invasive methods have been validated. The most promising is gargle lavage sampling. This non-invasive and painless self-sampling method can be performed without contact with health care personnel. Several gargle lavage sampling techniques have been tested to date [6–8], but none of them have used stabilization medium to preserve genomic and viral nucleic acid. In this study, a new collection medium was developed and validated for self-sampling and subsequent SARS-CoV-2 rRT-PCR testing. Moreover, gargle lavage samples obtained via this new sampling method were validated for antigen testing.”

  1. Provide statistical significance to your figures. 

Response: The requirement has been addressed. Statistical significance was provided to figures where it is relevant.

  1. Include the statistical significance of the diagnostic tests conducted. Also, add the number of replicates for each test or the number of times it was tested. 

Response: The requirement has been addressed.

  1. Discuss the reasons why the sensitivity of the test is very low? What contributes to this occurrence? 

Response: The requirements have been addressed. We added the explanation to the discussion part.

“Most antigen assays are designed to be used in combination with NPSs, alternatively, oropharyngeal swabs or saliva samples [29,30]. The combination of easily self-collected gargle samples and rapid SARS-CoV-2 antigen detection assays could enable improvements and new opportunities for containing the COVID-19 pandemic. Nevertheless, no antigen assay for SARS-CoV-2 antigen detection with GL are available. Although GL samples are comparable to NPSs for SARS-CoV-2 detection using rRT-PCR methods, they are too diluted for the antigen-based detection of SARS-CoV-2, which is less sensitive than rRT-PCR.” 

Round 2

Reviewer 1 Report

The authors updated the manuscript as suggested by the reviewers. I believe the manuscript in now suitable for publication.